# Antifungal Effects and Active Components of *Ligusticum* *chuanxiong*

**DOI:** 10.3390/molecules27144589

**Published:** 2022-07-19

**Authors:** Huabao Chen, Yingchun Zhao, Guangwei Qin, Yan Bi, Guizhou Yue, Min Zhang, Xiaoli Chang, Xiaoyan Qiu, Liya Luo, Chunping Yang

**Affiliations:** 1College of Agronomy, Sichuan Agricultural University, Chengdu 611130, China; chenhuabao12@163.com (H.C.); 18891307662@163.com (Y.Z.); qinguangwei163@163.com (G.Q.); 15982383506@163.com (Y.B.); yalanmin@126.com (M.Z.); xl_chang14042@sicau.edu.cn (X.C.); qxytzp@163.com (X.Q.); sicauluoliya@163.com (L.L.); 2College of Science, Sichuan Agricultural University, Ya’an 625000, China; yueguizhou@sicau.edu.cn

**Keywords:** *Ligusticum chuanxiong*, antifungal activity, Senkyunolide A, Ligustilide

## Abstract

The separation of chemical components from wild plants to develop new pesticides is a hot topic in current research. To evaluate the antimicrobial effects of metabolites of *Ligusticum* *chuanxiong* (CX), we systematically studied the antimicrobial activity of extracts of CX, and the active compounds were isolated, purified and structurally identified. The results of toxicity measurement showed that the extracts of CX had good biological activities against *Botry**tis cinerea*, *Sclerotinia sclerotiorum*, *Alternaria alternata* and *Pythium aphanidermatum*, and the value of EC_50_ were 130.95, 242.36, 332.73 and 307.29 mg/L, respectively. The results of in vivo determination showed that under the concentration of 1000 mg/L, the control effect of CX extract on *Blumeria graminis* was more than 40%, and the control effect on *Botry**tis cinerea* was 100%. The antifungal active components of CX were identified as Senkyunolide A and Ligustilide by mass spectrometry and nuclear magnetic resonance. The MIC (minimum inhibitory concentration) value of Senkyunolide A and Ligustilide against *Fusarium graminearum* were 7.81 and 62.25 mg/L, respectively. As a new botanical fungicide with a brightly exploitative prospect, CX extract has potential research value in the prevention and control of plant diseases.

## 1. Introduction

Since the 1980s, the research on plant resources with antimicrobial activity has been carried out at home and abroad. Scholars have conducted a lot of research on the effective components of plant antifungal and sterilization [1]. The functional components of many medicinal plants, such as volatile oils, alkaloids, flavonoids, organic acids, terpenoids, glycosides, quinones, esters, phenols and so on, have good antimicrobial effects [2]. It was found that 75% ethanol extract of Elsholtzia splendens had the most significant effect on *Botryodiplodia theobromae*, with an inhibition rate of 54.13% [3]. Cumin acid had good antimicrobial activity against a variety of pathogens, among which the virulence to Valsa mali was the strongest [4]. Carvanol had strong antimicrobial activity against *Colletotrichum gloeosporioides* and *Alternaria alternata*, with the values of IC_50_ being 40.89 μg/mL and 18.19 μg/mL, respectively [5]. Therefore, it is a valuable work to screen antimicrobial plants and identify their active components.

CX (named as Rhizoma chuanxiong before 2010 in the Chinese Pharmacopoeia), the dried rhizome of CX Hort., known as Chuan-Xiong (CX) in folk medicine, and belonging to the Umbelliferae family, is one of the oldest and most popular herbal medicines in the World. CX is warm in property and pungent in flavor, with the functions of activating qi, promoting blood circulation, expelling wind and alleviating pain, which has high medicinal value. It grows in a mild climate environment [6], and it is mainly produced in Guanxian Sichuan, Yunnan, Guizhou, Guangxi and other places. Its chemical composition includes phthalide and its dimer, alkaloids, organic acid phenols, polysaccharides, brain glycosides and ceramides. The active components of CX have various pharmacological activities on cardio-cerebrovascular system, nervous system, respiratory system and so on [7]. The acetone extract of CX had a good inhibitory effect on *Fusarium oxysporum*, *Botry**tis cinerea*, *poplar canker*, *Sclerotinia sclerotiorum* and *Fusarium graminearum* [8].

In preliminary research, it was determined that CX had broad-spectrum antifungal activity, but its antifungal activity components were not clear. Therefore, the purpose of this study is to clarify the fungicide active components and structure of CX. Then, extraction, silica gel column chromatography and high performance liquid chromatography (HPLC), and the data were analyzed by modern spectral techniques (EI-MS, ^1^H-NMR, ^13^C-NMR) to clarify the structure of the active components, which laid a foundation for further development and utilization of CX extract as a fungicide.

## 2. Results

### 2.1. Toxicity Determination of Extracts of CX

The extract of CX showed different degrees of inhibition on the mycelial growth of 10 different plant-pathogenic fungi (Table 1). Among them, the inhibitory effects of the extract on *Botry**tis cinerea*, *Sclerotinia sclerotiorum*, *Pythium aphanidermatum*, *Alternaria alternata* and *Didymella glomerata* were significant, and the EC_50_ values were 130.95, 242.36, 307.29, 332.73 and 470.53 mg/L, respectively. The extract had a poor inhibitory effect on *Fusarium graminearum*, *Colletotrichum gloeosporioides*, *Fusarium oxysporum*, *Fusarium lateritium* and *Phytophthora infestans*, and their EC_50_ values were in the range of 500–1000 mg/L.

### 2.2. Bioassay of Extracts of CX In Vivo

#### 2.2.1. Control Effect on *Blumeria graminis*

Different concentrations of extracts of CX had different effects on the control of *Blumeria graminis* (Table 2). The higher the concentration, the better the prevention and control effects were. Among them, the protective effect is better than the curative effect. Under the protective effect, the extract concentration of 4000 mg/L had the best control effect on *Blumeria graminis*.

#### 2.2.2. Control Effect on *Fusarium graminearum*

Different concentrations of extracts of CX had different degrees of control effects on *Fusarium graminearum* (Table 3). The higher the concentration, the better the control effect was, and the protective activity was better than the curative activity, but the control effect of the extract was lower than that of the control agent (86 mg/L Tebuconazole). When the concentration of extracts was 4000 mg/L, the protective and curative activities of *Fusarium graminearum* were the best. The RDCEs of the protective activities were 50.01%, 39.98% and 11.90%, respectively.

#### 2.2.3. Control Effect on *Botry**tis cinerea*

Different concentrations of extracts of CX had different degrees of control effects on *Botry**tis cinerea* (Table 4). When the concentration of the extract was 1000 mg/L, it had obvious protective and curative activities on *Botry**tis cinerea*. The RDCEs of the protective activities were 100.00%, and those of the curative activities were 100.00%, 75.00%, 45.47% at 3, 5 and 7 days after treatment, respectively. When the concentration of extract was 2000 mg/L, the RDCE of protective and curative activities reached 100%, which can achieve the control effect of the control agent (100 mg/L pyraclostrobin).

#### 2.2.4. Control Effect on *Colletotrichum gloeosporioides*

Different concentrations of CX extract have different degree of control effect on *Colletotrichum gloeosporioides* (Table 5). The higher the concentration, the better the control effect was, and among which the protective effect is better than the curative effect. Under the protective effect, the control effect was the best when the concentration of CX extract was 4000 mg/L, and the control effect was 48.04% after being treated for 7 days.

### 2.3. Tracking Results of Antifungal Activity of Different Fractions of CX Extract by Silica Gel Column Chromatography

Seven fractions of CX extracts were obtained, and their antifungal activities are shown in Table 6. Results showed that the fractions of FrD2 and FrD5 had antifungal activity. At the concentration of 400 mg/L, spore germination inhibition rates of FrD2 and FrD5 on *Botrytis cinerea* were 0.00%.

### 2.4. Separation and Purification and Tracking Results of Active Compounds by HPLC in CX Extract

The compounds CQ1-CQ5 were isolated and purified. The antifungal activity of the compounds were traced (Table 6). The results showed that CQ2 and CQ4 had obvious inhibitory effect on *Botrytis cinerea*. At the concentration of 62.25 mg/L, spore germination inhibition rates of CQ2 and CQ4 on *Botrytis cinerea* were 0.00%.

### 2.5. Chemical Structure Identification of Active Compounds from CX

#### 2.5.1. Compound CQ2

Compound CQ2 was identified as Yellowish oil. Its molecular formula is determined as C_12_H_16_O_2_ (M/Z 193.2[M+H]^+^) based on ESIMS data (M/Z 193.2[M+H]^+^), showing 5 degrees of unsaturated. ^1^H-NMR spectra (Table 7) showed two olefinic signals (1H, DT, J = 3.4, 8.9 Hz), 6.20 (1H, DT, J = 2.0, 9.7 Hz) and one mesine signal at X4.91 (1H, DD, J = 3.7,7.6 Hz). Amethyl group is in (3H, t, J = 7.1). ^13^C-NMR spectra (Table 8) show 12 carbon signals assigned to a Lactone carbonyl carbon (UNK C 171.4), four olefin atoms (UNK C 161.6, 128.5, 124.6, 116.9), One oxy-methine carbon (UNK C 82.6), five methyl groups (UNK C 32.0, 26.8, 22.5, 20.9, 22.4), and one methyl carbon (UNK C 14.0). These spectral data are the same as those released by Senkyunolide A [9,10]. Thus, the compound was identified as Senkyunolide A and its molecular formula was obtained (Figure 1).

#### 2.5.2. Compound CQ4

Compound CQ4 was identified as colorless oil. According to the HRESIMS data (M/Z 191.1051 [M+H]^+^), its molecular formula is determined as C_12_H_14_O_2_, indicating six degrees of unsaturated. ^1^H-NMR data (Table 6) show that δH is 6.24 (1H, D, J = 9.5Hz) 5.97 (1H, m) and 5.20 (1H, T, J = 8.0 Hz) for three olefin protons and one methyl proton (δH 0.93 (3H, T, J = 7.4 Hz)). ^13^C-NMR spectrum (Table 7) contains twelve carbon signals, one lactone carbonyl carbon (δC 167.8), six olefin carbons (δC 148.7, 147.2, 130.0, 124.1, 117.2, 113.1), four methylene carbons (δC 28.2, 22.5, 18.6, 18.6) and one methyl carbon (δC 13.9), respectively. These spectral data are consistent with the published data of Ligustilide [11,12]. Thus, compound CQ4 was identified as Ligustilide, and its molecular formula was obtained (Figure 2).

### 2.6. Antifungal Activity of Active Compounds Ingredients

Senkyunolide A and Ligustilide isolated from extracts of CX had inhibitory effects on the spore germination of *Fusarium graminearum*, *Botryis cinerea*, *Colletotrichum gloeosporioides* and *Fusarium oxysporum* (Table 8). The MIC values of Senkyunolide A against *Fusarium graminearum*, *Botryis cinerea*, *Colletotrichum gloeosporioides* and *Fusarium oxysporum* were 7.81 mg/L, 250 mg/L, 250 mg/L and 250 mg/L, respectively; the MIC values of Ligustilide effective against *Fusarium graminearum*, *Botryis cinerea*, *Colletotrichum gloeosporioides* and *Fusarium oxysporum* were 62.25 mg/L, 125 mg/L, 500 mg/L, and 250 mg/L, respectively.

## 3. Discussion

Plants can produce rich secondary metabolites. Statistics shows that more than 400,000 kinds of secondary metabolites that have significant biological activities against phytopathogenic fungi and bacteria have been found thus far. The phytochemical components of CX is complex, and more than 100 compounds such as haplophthalides, polyploides, alkaloids, terpenes, organic acids, lipids and polysaccharides have been isolated from it [13]. The use of these active natural products to develop new botanical fungicides has become an important research field.

In this study, the extract of CX was found that it has significant inhibitory effect on many plant pathogens, such as *Botryis cinerea*, *Sclerotinia sclerotiorum* and *Fusarium graminearum*. Among them, CX extract has the best control effect of on *Blumeria graminis* and *Botrytis cinerea*, and the inhibition rates (1000 mg/L) were more than 40% and 100%, respectively. Currently, studies of the medical and agricultural antimicrobial activity of CX extract have drawn more and more attention. In medicine, studies have reported that CX extract has significant antimicrobial effect on Escherichia coli, Salmonella, solanacearum, Staphylococcus albicans, Staphylococcus aureus and double-streptococcus pneumoniae [14]. In agriculture, studies have reported that CX extract has good antimicrobial effect on *Penicillium*, *Sclerotinia sclerotiorum* and *Fusarium graminearum*. For example, Xie et al. found that CX extract had a significant inhibitory effect on white mold and brown rot phytophthora and finger penicillium, which could reach 100% at concentrations of 1.0 × 10^−3^ g/mL [8,15]. Chen et al. found that the ethanol extract of CX had a good inhibitory effect on *Penicillium*, and the inhibiting rate was over 92.79% [16]. Tang et al., found that CX extract has antifungal effect on *Penicillium* citrus and the antifungal effect decreased with the decrease in the concentration of the extract. When the concentration was 500 mg/mL, the inhibition rate to Penicillium citrus was 100% [17]. Zhang et al. found that the inhibition rates of CX extract on the mycelial growth of *Sclerotinia sclerotiorum* and *Fusarium graminearum* were more than 60% [8].

Two active components were isolated and purified from the alcohol extract of CX Hort in this study, and they were identified as Senkyunolide A and Ligustilide, respectively. Senkyunolide A and Ligustilide are both phthalide compounds. The bioassay results showed that Senkyunolide A and Ligustilide had different inhibitory effects on *Fusarium graminearum*, *Botryis cinerea*, *Colletotrichum gloeosporioides* and *Fusarium oxysporum*. Among them, the MIC amounts of Senkyunolide A and Ligustilide against *Fusarium graminearum* were 7.81 mg/L and 62.25 mg/L. At present, the biological activities of these two compounds are mainly in medicine, and to a lesser extent in agriculture. For example, Senkyunolide A has good pharmacological effects such as anti-oxidative injury, anti-inflammation, anti-tumor and vasodilation [18]. Ligustilide has good pharmacological effects such as anti-atherosclerosis, anti-inflammation and analgesia, anti-senile dementia and anti-tumor [14]. In agriculture, Ligustilide has significant insecticidal activity against *Spodoptera litura* [19]. However, it has not been reported that these two compounds can inhibit plant pathogens such as *Fusarium graminearum*, *Botryis cinerea*. Therefore, the extract of CX as a plant fungicide will have potential applications.

The extract of CX had broad-spectrum antifungal activity *in vitro* and *in vivo*. The active components were isolated and purified by extraction, silica gel column chromatography and HPLC, and Senkyunolide A and Ligustilide further were identified by modern spectrum technology. The results of microporous plate method showed that Senkyunolide A and Ligustilide had an obvious inhibitory effect on *Fusarium graminearum*. However, the mechanism of Senkyunolide and Ligustilide inhibiting the growth and toxin production of wheat scab is not clear. Because *Fusarium asiaticum* is one of the main pathogens of wheat scab, it accumulates a large amount of single-ended mycotoxins such as nivalenol (NIV) in wheat grains [20]. Further, we will continue analyzing the effects of Senkyunolide and Ligustilide on the accumulation of NIV and the related indexes of mycelial cell membrane, cell wall and respiratory metabolism, and illustrate the mechanism of Senkyunolide and Ligustilide inhibiting the growth and NIV production of *Fusarium asiaticum* at the molecular level.

## 4. Materials and Methods

### 4.1. Materials

CX was provided by Sichuan Shushan Daogen Chinese Herbal Medicine Co., Ltd. Pathogens including *Fusarium graminearum*, *Blumeria graminis* (Bgt), *Botryis cinerea*, *Colletotrichum gloeosporioides*, *Sclerotinia sclerotiorum*, *Fusarium oxysporum*, *Fusarium lateritium*, *Alternaria alternata*, *Pythium aphanidermatum*, *Didymella glomerata* and *Phytophthora infestans* were provided by the College of Agronomy of Sichuan Agricultural University.

### 4.2. Preparation of Extracts of CX

Solvent extraction method was used [21,22]. The plant samples were dried in an electrothermal constant temperature blast drying box at 55 °C, and the dried plant powder was crushed by the grinder and sealed away from light in the fresh-keeping bag. The dried plant powder (2 kg) was soaked and extracted with 5 times volume of ethanol in dark. The extract was filtered, and then the same amount of ethanol was added after 2 days. The extract was filtered out after that and the two parts of the filtrate was combined. The extract was concentrated to the paste extract by rotary evaporator at 50–60 °C, scraped off, and stored in the refrigerator at 4 °C.

### 4.3. Toxicity Determination of Extracts of CX

Mycelial growth inhibition assay was used to evaluate the inhibitory effect of extracts of CX on fungi [23]. The extract of CX was dissolved with dimethyl sulfoxide and then diluted with sterile water to 2000, 1000, 500, 250, 125 and 62.5 mg/L. The solution (1 mL) was mixed with 9 mL Potato Dextrose Agar (PDA) medium (containing 1% streptomycin sulfate) and then poured into a sterile Petri dish (9 cm in diameter) to make a medium-filled plate. After the medium was solidified, an agar block (with a diameter of 0.5 cm) containing pathogenic fungi to be tested was placed in each medium plane, and the fungi-carrying side of the agar block containing fungi was placed onto the surface of the medium. Every concentration repeated three times, and double-distilled water (containing 0.1% Tween 80 and 2% DMSO) was used as the negative control. Then, pathogenic fungi were cultured at 25 °C. The diameter of colony growth was measured by the cross method after 2–7 days, and the inhibition rate was calculated. Then, the toxicity regression equation and EC_50_ value of the extract were obtained according to the probit analysis method for toxicity.
(1)$$\mathrm{Colony}\;\mathrm{growth}\;\mathrm{diameter}\;\left(\mathrm{mm}\right)=\mathrm{Average}\;\mathrm{of}\;\mathrm{measured}\;\mathrm{diameters}–5\left(\mathrm{diameter}\;\mathrm{of}\;\mathrm{agar}\;\mathrm{block}\right)$$
(2)$$\mathrm{Inhibition}\;\mathrm{rate}\;\mathrm{of}\;\mathrm{mycelium}\;\mathrm{growth}\;\left(\%\right)=\frac{\mathrm{CK}-\mathrm{PT}}{\mathrm{CK}}\times 100\;$$

CK: Control colony growth diameter; PT: Colony growth diameter of chemical treatment.

### 4.4. Bioassay of Extracts of CX In Vivo

#### 4.4.1. Control Effect on *Blumeria graminis*

The paste extract was dissolved in DMSO, and then prepared into 250 mg/L, 500 mg/L, 1000 mg/L, 2000 mg/L and 4000 mg/L solutions with double-distilled water (contain 0.1% Tween 80). Wheat seeds were planted in a glass tube (4 cm in diameter), and the tube was sealed with parafilm. Seeds were cultured to the three-leaf stage at 20 ± 1 °C and 60–70% humidity with a 16:8 h light/dark photoperiod [24]. Then, wheat leaves were sprayed with 250 mg/L, 500 mg/L and 1000 mg/L extract solutions with three replications at each concentration. Leaves were also sprayed with 100 mg/L prothioconazole as a positive control and with double-distilled water (contain 2% DMSO and 0.1% Tween 80) as a negative control. Each treatment was repeated three times. Fresh spores of Bgt were inoculated onto the plants by shaking over the foliage of the wheat seedlings [25]. The protective activity was determined by spray application of the extract solution first followed by inoculation with the pathogenic fungi 24 h later, and the curative activity was determined by inoculation with the pathogenic fungi first, followed by spray application of the extract solution 24 h later. The methods for detecting the protective and curative activities of the subsequent experiment were the same. Wheat was further cultured after treatment, and the disease incidence was investigated 7, 9 and 10 days after treatment. A disease classification grading scale was established, and the disease index and the relative disease control efficiency (RDCE) were calculated according to Standards of China “GB/T 17980-22-2000” [26].

#### 4.4.2. Control Effect on *Fusarium graminearum*

Wheat seeds were planted in pots (15 cm in diameter) and cultured to the blooming stage at 26 ± 2 °C and 65–75% humidity in the greenhouse. Then, wheat leaves were sprayed with 1000 mg/L and 4000 mg/L extract solutions with three replications at each concentration. Leaves were also sprayed with 100 mg/L tebuconazole as a positive control and with double-distilled water (contain 2% DMSO and 0.1% Tween 80) as a negative control. A spore suspension of *Fusarium graminearum* (10 μL) was injected wheat panicles with a microinjector. Spores were observed and counted on a hemocytometer under an optical microscope (4 × 10). The method for adjusting the spore suspension in the later experiment was the same. It is advisable to adjust the spore concentration to 80–100 spores per field. Wheat was further cultured after treatment, and the disease incidence was investigated 7, 9 and 10 days after treatment. A disease classification grading scale was established, and the disease index and the RDCE were calculated referring to Standards of China “NY/T 1464.15_2007” [27].

#### 4.4.3. Control Effect on *Botry**t**is cinerea*

Fresh strawberry fruits of uniform size were selected, soaked in 75% ethanol for 2 min, washed with double-distilled water and then dried. Then, fruits were sprayed with 500 mg/L, 1000 mg/L and 2000 mg/L extract solutions with three replications at each concentration. Leaves were also sprayed with 100 mg/L pyraclostrobin as a positive control and with double-distilled water (contain 2% DMSO and 0.1% Tween 80) as a negative control. The equator of the strawberry was pricked with a sterile needle of a 1 ml injector, a 2 mm wound was formed, and then a suspension of *Botry**t**is cinerea* was evenly spread on the fruit surface. The fruit was placed in a culture plate and stored at 25 °C and 95% humidity after treatment. The disease incidence was investigated 3, 5 and 7 days after treatment. The disease was divided into five grades according to the percentage of *Botrytis cinerea* diseased area to fruit surface area, and then the disease index and the RDCE were calculated [28].

#### 4.4.4. Control Effect on *Colletotrichum gloeosporioides*

Fruits of Jincheng orange with consistent appearance and no mechanical damage were selected, soaked in 75% ethanol for 2 min, washed with double-distilled water and then dried. Then, the fruits were sprayed with 1000 mg/L, 2000 mg/L, and 4000 mg/L extract solutions. Three biological replicates were performed at each concentration with 10 fruits per treatment. Leaves were also sprayed with 86 mg/L pyraclostrobin as a positive control and with double-distilled water (containing 2% DMSO and 0.1% Tween 80) as a negative control. *Colletotrichum gloeosporioides* was inoculated by needle puncture. The depth of the pinhole was 2 mm, and a spore suspension (10 μL) was dropped at the pinhole [29,30]. The fruits were cultured at 28 °C and 95% relative humidity after treatment. The diameters of the lesions were measured by the cross method, and the inhibition rate was calculated 7 days after treatment using the following formula:(3)$$\mathrm{Inhibition}\;\mathrm{rate}\;\left(\%\right)=\frac{\mathrm{CK}-\mathrm{PT}}{\mathrm{CK}}\times 100$$

CK: Control colony growth diameter; PT: Colony growth diameter of chemical treatment.

#### 4.4.5. Data Processing and Analysis

SPSS.23 software was used for regression analysis, variance analysis and multiple comparison of the test data.

### 4.5. Isolation, Purification and Structure Identification of Active Compounds from CX

#### 4.5.1. Macroporous Resin Separation

The D101 macroporous resin was soaked in absolute ethanol for 24 h. After the resin was fully swollen, it was loaded into a chromatography column and rinsed repeatedly with absolute ethanol until the supernatant was free of white turbidity. Then, the resin was rinsed with distilled water until there was no ethanol. After 10 L of the treated CX extract was statically adsorbed with 2 kg D101 macroporous resin for 48 h, the D101 macroporous resin was loaded into the chromatographic column, first eluted with 3 times the column volume of distilled water, and the eluent was discarded; then, 5 Elution with 70% ethanol was used, and the filtrate was collected and concentrated under reduced pressure to obtain the CX extract.

#### 4.5.2. Petroleum Ether Extraction and Silica Gel Column Chromatography

The CX extract was evenly dispersed with water, and the aqueous solution of the CX extract was extracted four times with an equal volume of petroleum ether, and the organic phase was concentrated under reduced pressure at 45 °C to obtain the petroleum ether extract. The petroleum ether extract was separated by column chromatography using normal phase silica gel, and the elution solvent was successively eluted with different mixtures of chloroform and methanol (1:0, 3:1 and 0:1, *v/v*), and the fractions were collected. The extract was separated into 7 fractions (Fr. D1-D7) by silica gel column chromatography. Further separation is performed by silica gel column chromatography: using petroleum ether:ethyl acetate (5:1-2:1) as eluent, elute Fr. D2 to obtain fraction Fr. D2-1; using petroleum ether:acetone (5:1-2:1) as the eluent, elute Fr. D5 to obtain fraction Fr. D5-1. The antimicrobial active fractions were traced and selected by the spore germination method using *Botry**t**is cinerea* as an indicator pathogen.

#### 4.5.3. HPLC Preparation

The components (FR. D2-1, FR. D5-1) are fitted with Phenomenex C18 columns (250 × 10 mm, Phenomenex; Aschaffenburg, Germany) and UV detector. The fr. D2-1 component was eluted with 30% (*v/v*) acetonitrile containing 0.1% formic acid to obtain compound numbers CQC1 (20 mg), CQC2 (20 mg) and CQC3 (5 mg). The fr. D5-1 component was eluted with 35% (*v/v*) acetonitrile containing 0.1% formic acid to obtain compound codes CQC4 (20 mg) and CQC5 (5 mg). The antimicrobial active compound was traced and selected by spore germination method with *Botry**t**is cinerea* as indicator pathogen.

#### 4.5.4. Compound Structure Identification

The purity was determined by HPLC. The mass spectrum, ^1^H-NMR (nuclear magnetic resonance) and ^13^C-NMR spectra of the compounds with higher purity were determined (the ^1^H-NMR spectra were 400 MHz, the ^13^C-NMR spectra were 100 MHz and the solvent was CDCl_3_). The chemical structure of the compound was identified according to spectroscopic data.

#### 4.5.5. Determination of Compound Activity

The MIC against the pathogen spores was determined by the microtiter method [26]. Dimethyl sulphoxide was used to prepare the pure compound into 20.00 mg/L mother liquor, and the mother liquor was diluted into 1000 mg/L, 500 mg/L, 250 mg/L, 125 mg/L, 62.25 mg/L, 31.13 mg/L, 15.63 mg/L, 7.813 mg/L, 3.91 mg/L, and 1.95 mg/L aqueous solution with sterile water by double-dilution method. Add 5μL of compound solution of different concentrations to each single well of 96-well plate, add 45 μL of medium at 50 °C to each well, mix thoroughly, prepare the medium containing drug with the desired concentration, stand for 10 min, add 10 μL of spore suspension to each well, and add the same amount of dimethyl sulphoxide solution to the single well of 96-well plate as control. Each treatment was repeated 3 times. After incubation at 28 °C, for 48 h, the growth of mycelium in all pore plates was observed under microscope, and the concentration of no mycelium growth was the minimum inhibitory concentration.

## Figures and Tables

**Figure 1 molecules-27-04589-f001:**
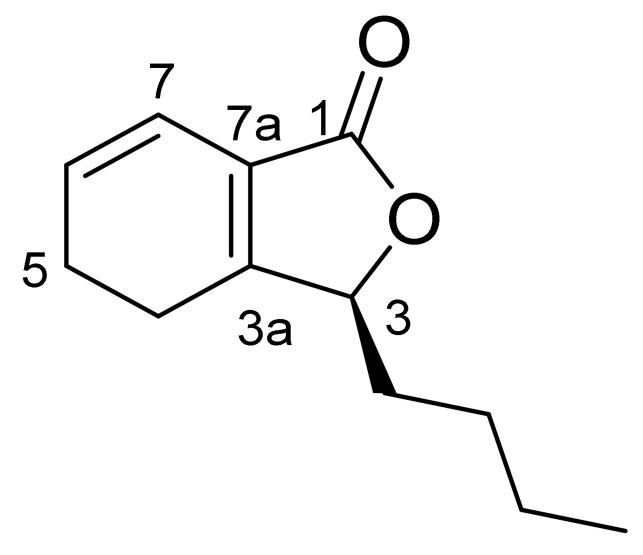
Structural formula of Senkyunolide A.

**Figure 2 molecules-27-04589-f002:**
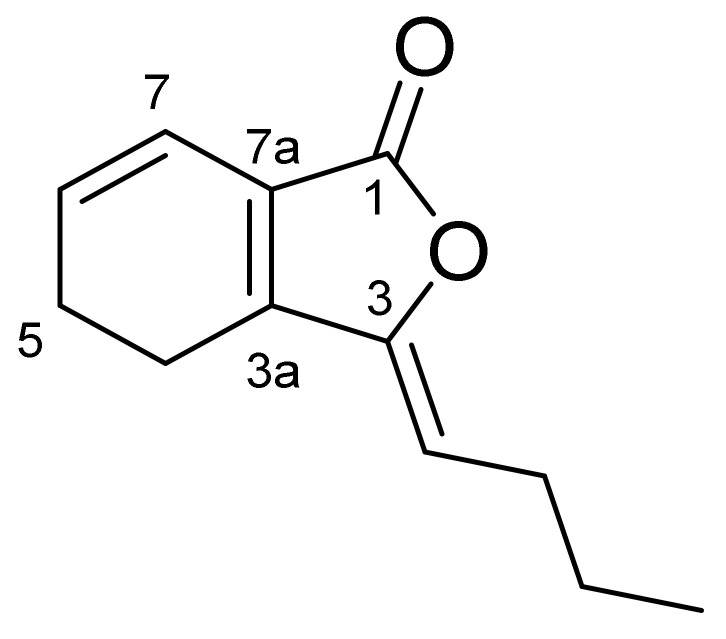
Structural formula of Ligustilide.

**Table 1 molecules-27-04589-t001:** Toxicity test results of CX plant extracts against 10 kinds of plant pathogens.

Plant Pathogen	Toxicity Regression Equation	Correlation Coefficient	EC_50_ (mg/L)	Confidence Interval (mg/L)
*Fusarium graminearum*	y = 1.1737x + 5.0210	0.9800	959.64	(670,1370)
*Botry* *tis cinerea*	y = 4.0417x + 8.5684	0.9909	130.95	(110,160)
*Colletotrichum gloeosporioides*	y = 0.8628x + 5.1074	0.8606	750.78	(310,1810)
*Sclerotinia sclerotiorum*	y = 2.8467x + 6.7523	0.9536	242.36	(210,280)
*Fusarium oxysporum*	y = 1.2462x + 5.1013	0.9804	826.49	(570,1210)
*Fusarium lateritium*	y = 0.8047x + 5.2308	0.9180	516.55	(310,850)
*Alternaria alternata*	y = 2.1608x + 6.0327	0.9624	332.73	(270,420)
*Pythium aphanidermatum*	y = 0.8966x + 5.4595	0.9471	307.29	(130,720)
*Didymella glomerata*	y = 1.5346x + 5.5024	0.9385	470.53	(300,750)
*Phytophthora infestans*	y = 1.1329x + 5.3577	0.9810	534.05	(370,770)

**Table 2 molecules-27-04589-t002:** Control effect of CX extract on *Blumeria graminis* by pot trial.

Treatment	Concentration	Protective Activity			Curative Activity		
	(mg/L)	7 d (%)	9 d (%)	11 d (%)	7 d (%)	9 d (%)	11 d (%)
Extracts of CX	4000	68.45 b	64.71 b	53.37 b	55.77 b	42.31 b	36.64 b
	2000	66.71 b	62.97 b	51.42 b	22.00 c	21.37 c	15.52 c
	1000	52.46 c	50.50 c	49.15 b	19.85 cd	15.38 d	11.62 d
	500	48.39 cd	47.37 cd	48.54 b	15.79 de	12.55 e	4.31 e
	250	46.10 d	43.86 d	42.34 c	14.98 e	10.46 e	1.71 ef
Prothioconazole	100	92.31 a	84.53 a	80.32 a	88.62 a	76.35 a	70.46 a

Statistical significance was determined using one-way ANOVA. Different lowercase letters in the footnote in the same column show significant difference in the relative disease control efficiency of different treatment at each time points (*p* < 0.05).

**Table 3 molecules-27-04589-t003:** Control effect of CX extract on *Fusarium graminearum* by pot trial.

Treatment	Concentration	Protective Effect			Curative Effect		
	(mg/L)	7 d (%)	9 d (%)	11 d (%)	7 d (%)	9 d (%)	11 d (%)
CX	1000	31.26 c	9.12 c	7.14 c	25.01 c	3.96 c	2.38 c
	4000	50.01 b	39.98 b	11.90 b	36.39 b	32.82 b	7.14 b
Tebuconazole	86	71.88 a	81.99 a	88.10 a	54.56 a	87.50 a	80.95 a

Statistical significance was determined using one-way ANOVA. Different lowercase letters in the footnote in the same column show significant difference in the relative disease control efficiency of different treatment at each time points (*p* < 0.05).

**Table 4 molecules-27-04589-t004:** *In vivo* control effect of CX plant extracts on *Botry**tis cinerea*.

Treatment	Concentration	Protective Effect			Curative Effect		
	(mg/L)	3 d (%)	5 d (%)	7 d (%)	3 d (%)	5 d (%)	7 d (%)
	2000	100.00 a	100.00 a	100.00 a	100.00 a	100.00 a	100.00 a
CX	1000	100.00 a	100.00 a	100.00 a	100.00 a	75.00 b	45.47 b
	500	100.00 a	62.51 b	55.56 b	100.00 a	57.13 c	9.11 c
Pyraclostrobin	100	100.00 a	100.00 a	100.00 a	100.00 a	100.00 a	100.00 a

Statistical significance was determined using one-way ANOVA. Different lowercase letters in the footnote in the same column show significant difference in the relative disease control efficiency of different treatment at each time points (*p* < 0.05).

**Table 5 molecules-27-04589-t005:** *In vivo* control effect of CX plant extracts on *Colletotrichum gloeosporioides*.

Treatment	Protective Effect			Curative Al Effect	
	Concentration (mg/L)	Average Diameters of Lesions (cm)	Inhibitory Rate (%)	Mean Diameter of Spot (cm)	Inhibitory Rate (%)
CX	4000	0.83	48.04 b	0.93	41.67 b
	2000	1.03	39.22 c	1.18	26.04 c
	1000	1.28	24.51 d	1.47	8.33 d
Tebuconazole	86	0.52	69.61 a	0.54	66.15 a
CK	-	1.70	-	1.60	-

CK is negative control. Statistical significance was determined using one-way ANOVA. Different lowercase letters in the footnote in the same column show significant difference in the inhibitory rate (*p* < 0.05).

**Table 6 molecules-27-04589-t006:** Spore germination inhibition rate of 7 fractions of CX extract obtained by silica gel column chromatography and 5 fractions obtained by HPLC on *Botrytis cinerea*.

Samples		Concentration (mg/L)	Spore Germination Inhibition Rate (%)
silica gel column chromatography	FrD1	400	72.35
	FrD2		0.00
	FrD3		74.69
	FrD4		22.82
	FrD5		0.00
	FrD6		74.04
	FrD7		21.56
HPLC	CQ1	62.25	92.85
	CQ2		0.00
	CQ3		93.64
	CQ4		0.00
	CQ5		95.86

**Table 7 molecules-27-04589-t007:** ^1^H and ^13^C NMR data of CQ2 and CQ4 (CDCl_3_, δ in ppm, J in Hz).

No.	CQ2 ^a^		CQ4 ^a^	
	*δ* _H_	*δ* _C_	*δ* _H_	*δ* _C_
1		171.4		167.8
2				/
3	4.91 (dd, 3.7, 7.6)	82.6		148.7
3a		161.6		124.1
4	2.44 (m)	20.9	2.57 (t, 9.4)	18.6
5	2.44 (m)	22.5	2.43 (m)	18.6
6	5.89 (dt, 3.4, 8.9)	128.5	5.97 (m)	130.0
7	6.20 (dt, 2.0, 9.7)	116.9	6.24 (d, 9.5)	117.2
7a		124.6		147.2
8	1.86 (m); 1.51 (m)	32.0	5.20 (t, 8.0)	113.1
9	1.35 (m)	26.8	2.34 (q, 7.6)	28.2
10	1.35 (m)	22.4	1.47 (m)	22.5
11	0.89 (t, 7.1)	14.0	0.93 (t, 7.4)	13.9

a: Recorded at 400 MHz for ^1^H and 100 MHz for ^13^C. dd: doublet of doublets, m: multiplet, dt: double triple peak, t: triple peak, d: double peak, q: quadruple peak

**Table 8 molecules-27-04589-t008:** Determination results of the MIC of compounds CQ2 and CQ4 against the four pathogenic fungi.

Treatments	MIC (mg/L)			
	*Fusarium* *graminearum*	*Botryis* *cinerea*	*Colletotrichum* *gloeosporioides*	*Fusarium* *oxysporum*
Senkyunolide A	7.81	250	250	250
Ligustilide	62.25	125	500	250
pyraclostrobine	40	2	1.25	2.5

## Data Availability

Not applicable.

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
