# Peer review of "Antifungal Effects and Active Components of Ligusticum chuanxiong"

_molecules, 2022, doi:10.3390/molecules27144589_

Round 1
Reviewer 1 Report
The article need improvement, in many places the datas are not in order. The biological activity need clarity...If you are doing the activity with isolated compound, column procedure can be mentioned first before biological activity. Are the isolated compounds activity checked ? Graphical representation is needed..Results need to be arranged in a proper way.Graphs should have clarity in all aspects. The novelty need to be projected with what is reported and what is not reported for the topic.
Author Response
Dear Editors and Reviewers:
感谢您的来信和审稿人对我们题为“Ligusticum的抗真菌作用和活性成分”的手稿的评论。传雄“(ID:分子-1788659)。这些意见对于修改和改进我们的论文都是有价值的,非常有帮助,对我们的研究也具有重要的指导意义。我们仔细研究了意见,并作了更正,希望得到批准。修改后的部分在论文中标有红色。论文中的主要更正和对审稿人评论的回应同样流畅:
回复审阅者的评论:
评论者 #1:
- 回复评论:文章需要改进,在很多地方数据不合规定。
回复:感谢您的建议,作者已经进行了更正。
- 回应评论:生物活性需要澄清...如果您使用分离的化合物进行活性,则可以在生物活性之前首先提及列程序。
回应:考虑到审稿人的建议,我们添加了硅胶柱层析各组分抗真菌活性的跟踪结果,川雄提取物活性成分的分离纯化和抗真菌活性的跟踪结果。有关更多详细信息,请参见第 2.3 节。和 2.4.
- 对评论的回应:结果需要以适当的方式安排。图表应该在各个方面都清晰明了。
回应:我们根据审稿人的建议重写了这部分。
对评论的回应:新颖性需要用报道的内容和未报道的主题来预测。
回应:考虑到审稿人的建议,我们在讨论部分增加了蜀香川雄提取物的研究报告。更多细节见第3节第二和第三段。
我们尽最大努力改进稿件,并对稿件进行了一些修改。这些变化不会影响论文的内容和框架。
我们衷心感谢编辑/审稿人的热情工作,并希望更正得到认可。
再次感谢您的意见和建议。

Reviewer 2 Report
The article entitled “Antifungal effects and active components of Ligusticum. chuanxiong” is a study that had the aim to detect some of the antimicrobial effects of CX extracts and of two isolated, purified, identified, active components of CX, called Senkyunolide A and ligustilide. Given the favorable effects that these two substances have determined, the authors believe that they could have a role in the prevention and control of myco- and phytopathogens of plants.
I would provide some suggestions with the aim to ameliorate the work.
In the Abstract:
Botryis cinerea should be Botrytis cinerea
In the Introduction:
At line 5: “antimicrobial and antimicrobial” the word is repeated twice.
At the end of the introduction the aim of the study is not clearly and exhaustively specified and what type of analyzes the authors intend to carry out.
In the Results:
The two formulas of “Inhibition rate” do not have the legend.
The Discussion should be remodulated. Briefly, the Discussion paragraph should start with a short summary of the findings and of the study and then moving on to the importance of the study and the results obtained. Unfortunately, in the paragraph these elements do not appear, so they should be introduced.
In the same way, at the end of the paragraph, it is necessary to provide a conclusion indicating what could be the future developments of the topic and why it would be important to continue these studies.
In general:
I would suggest indicating fungi names with the genus in full.
The manuscript needs a revision of the English language due to the presence of some typing and punctuation errors.
Author Response
Dear Editors and Reviewers:
Thank you for your letter and for the reviewers’ comments concerning our manuscript entitled “Antifungal effects and active components of Ligusticum. chuanxiong” (ID: molecules-1788659). Those comments are all valuable and very helpful for revising and improving our paper, as well as the important guiding significance to our researches. We have studied comments carefully and have made correction which we hope meet with approval. Revised portion are marked in red in the paper. The main corrections in the paper and the responds to the reviewer’s comments are as flowing:
Responds to the reviewer’s comments:
Reviewer #2:
- Response to comment: In the Abstract:Botryis cinerea should be Botrytis cinerea.
Response: At line 9, modified the Latin form of Botrytis cinerea.
2.Response to comment: In the Introduction: At line 5:“antimicrobial and antimicrobial” the word is repeated twice.
Response: The repeated word in the Introduction were deleted.
3.Response to comment: At the end of the introduction the aim of the study is not clearly and exhaustively specified and what type of analyzes the authors intend to carry out.
Response: We have re-written this part according to the Reviewer’s suggestion. More details are provided in the third paragraph of the introduction.
- Response to comment: The two formulas of“Inhibition rate” do not have the legend.
Response: We are very sorry for our negligence of these and add to the legend.
- Response to comment: The Discussion should be remodulated. Briefly, the Discussion paragraph should start with a short summary of the findings and of the study and then moving on to the importance of the study and the results obtained.Unfortunately, in the paragraph these elements do not appear, so they should be introduced.
Response: Considering the Reviewer’s suggestion, We have recalibrated the discussion to supplement the research on secondary metabolites of CX, the importance of CX as a wide range of antibacterial plant-derived pesticide research, and the antibacterial effect of the active compound isolated from CX extract. More details in Section 3.
- Response to comment: In the same way, at the end of the paragraph, it is necessary to provide a conclusion indicating what could be the future developments of the topic and why it would be important to continue these studies.
Response: We have made correction according to the Reviewer’s comments.
- Response to comment: The manuscript needs a revision of the English language due to the presence of some typing and punctuation errors.
Response: Thanks for the suggestion , I carefully checked the manuscript for some typing and punctuation errors.
We tried our best to improve the manuscript and made some changes in the manuscript. These changes will not influence the content and framework of the paper.
We appreciate for Editors/Reviewers’ warm work earnestly, and hope that the correction will meet with approval.
Once again, thank you very much for your comments and suggestions.

Round 2
Reviewer 2 Report
Dear Author/s,
the changes made to the manuscript responded correctly to the observations.
A single note on the Discussions: what has been added is appreciable and must be inserted, but it is necessary that a small summary of what has been done and the results obtained to be discussed must always be done at the beginning of each discussion.
In fact, it is from these results that the discussion then starts, aimed, in general, at strengthening one's own results on the basis of other papers reported in references, or at explaining the reasons why one's results are not compatible with those found in references. So, I would suggest making a brief summary of the results and possibly the modalities that will be discussed shortly thereafter.
Best regards.
Author Response
Dear Editors and Reviewers:
Thank you for your letter and for the reviewers’ comments concerning our manuscript entitled “Antifungal effects and active components of Ligusticum. chuanxiong” (ID: molecules-1788659). Those comments are all valuable and very helpful for revising and improving our paper, as well as the important guiding significance to our researches. We have studied comments carefully and have made correction which we hope meet with approval. Revised portion are marked in red in the paper. The main corrections in the paper and the responds to the reviewer’s comments are as flowing:
Responds to the reviewer’s comments:
Reviewer:
Response to comment:A single note on the Discussions: what has been added is appreciable and must be inserted, but it is necessary that a small summary of what has been done and the results obtained to be discussed must always be done at the beginning of each discussion.
In fact, it is from these results that the discussion then starts, aimed, in general, at strengthening one's own results on the basis of other papers reported in references, or at explaining the reasons why one's results are not compatible with those found in references. So, I would suggest making a brief summary of the results and possibly the modalities that will be discussed shortly thereafter.
Response:We have re-written this part according to the Reviewer’s suggestion. We complement the emphasis on the results in other papers
and provide a brief summary of the results.
We appreciate for Editors/Reviewers’ warm work earnestly, and hope that the correction will meet with approval.
Once again, thank you very much for your comments and suggestions.
